# Autoimmune Polyendocrinopathy–Candidiasis–Ectodermal Dystrophy in Two Siblings: Same Mutations but Very Different Phenotypes

**DOI:** 10.3390/genes12020169

**Published:** 2021-01-26

**Authors:** Andrea Carpino, Raffaele Buganza, Patrizia Matarazzo, Gerdi Tuli, Michele Pinon, Pier Luigi Calvo, Davide Montin, Francesco Licciardi, Luisa De Sanctis

**Affiliations:** 1Postgraduate School of Pediatrics, University of Turin, 10126 Turin, Italy; 2Pediatric Endocrinology Unit, Regina Margherita Children’s Hospital, 10126 Turin, Italy; raffaele.buganza@unito.it (R.B.); gerdi.tuli@unito.it (G.T.); patrizia.matarazzo@unito.it (P.M.); luisa.desanctis@unito.it (L.D.S.); 3Pediatric Gastroenterology Unit, Regina Margherita Children’s Hospital, 10126 Turin, Italy; pcalvo@cittadellasalute.to.it (P.L.C.); michele.pinon@gmail.com (M.P.); 4Pediatric Immunology and Rheumatology, Regina Margherita Children’s Hospital, 10126 Turin, Italy; francesco.licciardi@gmail.com (F.L.); davide.montin@gmail.com (D.M.); 5Department of Public Health and Pediatric Sciences, University of Turin, 10126 Turin, Italy

**Keywords:** autoimmune polyendocrinopathy–candidiasis–ectodermal dystrophy (APECED), *AIRE* gene, genotype–phenotype correlation

## Abstract

Autoimmune polyendocrinopathy–candidiasis–ectodermal dystrophy (APECED), caused by mutations in the *AIRE* gene, is mainly characterized by the triad of hypoparathyroidism, primary adrenocortical insufficiency and chronic mucocutaneous candidiasis, but can include many other manifestations, with no currently clear genotype–phenotype correlation. We present the clinical features of two siblings, a male and a female, with the same mutations in the *AIRE* gene associated with two very different phenotypes. Interestingly, the brother recently experienced COVID-19 infection with pneumonia, complicated by hypertension, hypokalemia and hypercalcemia. Although APECED is a monogenic disease, its expressiveness can be extremely different. In addition to the genetic basis, epigenetic and environmental factors might influence the phenotypic expression, although their exact role remains to be elucidated.

## 1. Background

Autoimmune polyendocrinopathy–candidiasis–ectodermal dystrophy (APECED) is a rare monogenic disease, caused by mutations in the autoimmune regulator (*AIRE*) gene on chromosome 21 (21q22.3).

APECED is mainly known for the triad of hypoparathyroidism, primary adrenocortical insufficiency and chronic mucocutaneous candidiasis. The definite diagnosis of APECED requires one of the following three criteria: the presence of at least two out the three major clinical features or only one of the major components if a sibling has definite APECED or a disease-causing mutation in both alleles of the *AIRE* gene. [1]. However, many other manifestations have been described, including other endocrinopathy (hypergonadotropic hypogonadism, type 1 diabetes, autoimmune thyroid diseases, pituitary defects), gastrointestinal manifestations (autoimmune gastritis, autoimmune hepatitis, intestinal malabsorption, celiac disease), ectodermal abnormalities (keratitis, enamel dysplasia, vitiligo, alopecia, nail dystrophy) and others [1,2].

The major mutation found in Finns, i.e., the c.769C>T variant, and the 13 bp deletion in exon 8 (c.967–979del13), are the most common ones worldwide [1]. Currently, more than 100 pathogenic mutations have been described and recently new genetic variations have been added [3,4,5]. The *AIRE* gene plays an essential role in central tolerance. Mutations in the *AIRE* gene prevent the elimination of self-reactive T cells at the central level and induce a T regulatory cell (Treg) defect at peripheral levels, leading to the development of multiple autoimmune diseases at a young age [6,7].

Although APECED is the paradigm of a monogenic autoimmune disorder, it is characterized by a wide variability of the clinical expression. In the majority of studies, no genotype–phenotype correlation has been found, while several observations suggest that a partial genotype–phenotype correlation may exist for specific traits [6]. Therefore, clinical manifestations of APECED patients can vary, suggesting that other factors, such as environment, lifestyle, habits or other genetic mechanisms, might modulate the disease presentation [8]. In particular, disease-modifying genetic variants in APECED patients were recently supported by different findings in mice [9,10]. Recently, dominant-negative *AIRE* mutations involving the SAND or PHD-1 domains have been identified in patients presenting with much less severe APECED-like disease, indicating possible modulation of the phenotypic expression of common organ-specific autoimmune diseases [11].

Furthermore, the recent finding of important enhancer elements activated by nuclear factor κB (NF-κB) signaling that is necessary for *AIRE* gene expression, as well as the identification of further additional regulators, suggest that *AIRE* gene expression and its function are tightly regulated [11].

APECED syndrome could be potentially underdiagnosed or misdiagnosed due to its rarity, but also because of the variability in its presentation.

We report the case of two siblings with the same mutations in the *AIRE* gene but with very different phenotypes.

## 2. Case Reports

The girl (patient 1) in the first year of life developed oral candidiasis, onychodystrophy, transient transaminitis and hepatomegaly (with antinuclear antibodies -ANA- and actin smooth muscle antibodies -ASMA- positivity) and recurrent episodes of fever of unknown origin. At 1.6 years old, she presented a hypocalcemic tetanic crisis. The hematological exams were consistent with hypoparathyroidism, i.e., low blood calcium (1.42 mmol/L, normal range 2.3–2.8), high phosphorus (2.7 mmol/L, normal range 1.5–2) and low parathyroid hormone (PTH) (7.9 pg/mL, normal range 15–57). APECED was suspected and then confirmed by Sanger sequencing, which detected two compound heterozygous mutations in the *AIRE* gene, in exon 2 and exon 8, namely, NM_000383.4(*AIRE*):c.260T>C (p.Leu87Pro) and NM_000383.4(*AIRE*):c.967_979del (p.Leu323fs), respectively. Parental analysis showed that one mutation was inherited from the father and the other from the mother; by considering the absence of any signs or symptoms, no further laboratory investigations were performed.

Thereafter, she developed vitiligo, alopecia, enamel hypoplasia and autoimmune insulitis with positive glutamic acid decarboxylase (GADA) antibodies but no glycemic alteration, abdominal pain and bilateral swelling of the feet without radiographic abnormalities.

At 3.4 years old, for recurrent episodes of tetanic crisis and difficulty to maintain adequate serum calcium levels, notwithstanding the chronic therapy with calcium and calcitriol, she was enrolled in an experimental trial with recombinant human parathyroid hormone (rhPTH) (1–34). At 10 years of age, her height was 131.5 cm (10–25th percentile) with target height of 171 cm (90–97th percentile) and reduced growth velocity (3.3 cm/year, <3rd percentile) according to Tanner growth charts [12]; however, the dynamic growth hormone (GH) testing with Arginine did not show GH deficiency (GH values after arginine test of 11.4 ng/mL). At the age of 12, her abdominal pain worsened with chronic diarrhea and fat-soluble vitamin deficiencies were disclosed (vitamin A 0.21 mcg/mL, 25OH-vitamin D 14.6 ng/mL). The esophagogastroduodenoscopy (EGDS) and colonoscopy revealed macroscopical normal mucosa except for nodular duodenum; histological evaluation detected mild inflammatory infiltrate and voluminous lymphatic follicles in the duodenal bulb and in the rectum; immunohistochemistry of small and large bowel samples with chromogranin A staining showed the complete absence of enteroendocrine cells (EECs) in the stomach, duodenum and ileum, severe decrement of EECs in the ascending colon and normal expression in the rectum. From 13.6 years old, an estradiol-depot patch was placed for hypogonadism (estradiol < 5 pg/mL, anti-müllerian hormone 0.5 ng/mL, basal LH and FSH 0.4 U/L and 3.3 U/L, respectively, LH and FSH values after LHRH test 9.5 U/L and 18.8 U/L, respectively) Currently, patient 1 is 14 years old with height 146.7 cm (<3° percentile), but with catch-up growth (high velocity of 6.5 cm/year, > 97th percentile). Recently, she had SARS-CoV-2 asymptomatic infection.

The boy (patient 2) was evaluated for the first time at 6.4 years old, after his sister’s diagnosis. At the time of diagnosis, he had hypoparathyroidism with asymptomatic mild hypocalcemia, i.e., low blood calcium (1.07 mmol/L, normal range 2.3–2.8), high phosphorus (5.67 mmol/L, normal range 1.5–2) and low PTH (3 pg/mL, normal range 15–57), alopecia and mild oral candidiasis. The same compound heterozygous mutations in the *AIRE* gene identified in his sister were then detected. At 8.3 years old, for repeated episodes of hypocalcemic tetanic crisis despite therapy with calcium and calcitriol, rhPTH (1–34) treatment was started. Dermatitis rosaceiforme and severe acne were also present and a GADA-positive autoimmune insulitis, without glucose impairment, was then discovered. At 13.9 years old, hyponatremia linked to autoimmune adrenal insufficiency was detected during routine exams and therapy with hydrocortisone and fludrocortisone was started. Over time, he has shown regular growth and pubertal development: at the last evaluation at 19.8 years old, his height was 176.9 cm (50–75th percentile). At 19 years old, he developed COVID-19 pneumonia, treated with oxygen therapy through a nasal cannula, methylprednisolone (starting at 60 mg on the first day then down-titrated in the following days), antibiotics (azthromycin and ceftriaxone) and remdesivir. During the hospitalization, despite the discontinuation of fludrocortisone and the steroid tapering, he showed persistent hypertension (up to 200/120 mmHg) and hypokalemia. Antihypertensive therapy was started (with ramipril 10 mg/die and then, after discharge, amlodipine 10 mg/die and nebivolol 5 mg/die) with mild improvement of pressure values. Potassium chloride per os was added with subsequent normalization of serum potassium levels. He also developed hypercalcemia and hypercalciuria, requiring adjustments of treatment with calcium, calcitriol and rhPTH (1–34).

The main features of both patients are summarized in Table 1.

## 3. Discussion

The *AIRE* gene, pathogenetic for APECED, is expressed in thymic medullary epithelial cells where it induces the expression of a wide repertoire of peripheral tissue antigens that play a crucial role in central tolerance for the correct development of self-tolerance. Thus, the absence of *AIRE* expression results in an impaired clonal deletion of self-reactive thymocytes. Furthermore, there is now strong evidence for *AIRE* expression in peripheral tissues even though these levels are significantly lower than in thymic stromal cells [6].

The patients described here have two compound heterozygous mutations in the *AIRE* gene and one of these variants (i.e., 967–979 del13) is one of the most common in the literature, with variable prevalence in different populations of 71%, 53% and 48% in British, North American and Norwegian series, respectively [13,14,15].

The second mutation, NM_000383.4(*AIRE*):c.260T>C (p.Leu87Pro), has previously been described in a pediatric compound heterozygous patient with maternal inheritance, and paternal inheritance of the p.Leu323fs variant. The subject had a clinical diagnosis of APECED, with anti-omega antibodies and immunologic deregulation since birth. For the associated phenotype and the location within a region of the protein in which other single nucleotide variants produce lack of function alleles inhibiting proper dimerization of the AIRE protein, the mutation was classified as likely pathogenetic [16].

APECED is a recessive monogenic condition although, in a few cases, only one mutant allele of the *AIRE* gene has been reported in typical APECED patients, suggesting that the second mutation might be located in the regulatory regions of the gene [17].

As for the clinical features, chronic mucocutaneous candidiasis is usually the first of the main components to appear, even if APECED includes other ectodermal anomalies, as reported in the two cases presented, with different characteristics. Alopecia, which is usually localized (alopecia areata), as was seen in patient 1, can spread to the whole scalp (total alopecia), as it did in patient 2.

Hypoparathyroidism is the second most common major component of APECED and usually the first endocrine manifestation [1]. It should present with severe symptoms, as in patient 1, who had tetanic crisis. Treatment with oral calcium and vitamin D analogs represents the conventional treatments in children but they have to be modified frequently, based on body weight and intestinal absorption, and because they can lead to nephrocalcinosis due to hypercalciuria [18]. RhPTH (1–34) is an “off-label” treatment in pediatric ages, but in some clinical studies, it has allowed the maintenance of adequate levels of calcium and phosphate in the blood, to normalize urinary calcium excretion and to reduce tetanic episodes [18], as in our patients who needed high doses of oral calcium before RhPTH (1–34) treatment, and in which, particularly in patient 1, that treatment also led to a reduction of hypercalciuria. Its use over time in patient 1 was further justified by the important malabsorptive syndrome, which not only prevented the maintenance of plasma calcium values in the normal range, but also reduced growth.

The growth impairment reported in patient 1 can be determined by many causes, such as pubertal delay, malabsorption and chronic disease. It is noteworthy that hypogonadism was present only in patient 1, the female, thus confirming the different gender prevalence of this manifestation in APECED, with higher frequency in females [19]. Premature ovarian insufficiency develops in most patients with APECED, often before or shortly after menarche. In a recent study, it developed in 70% of patients at a median age of 16.0 years and in 71% of them before reaching adult height; therefore, timely initiation of hormone replacement therapy is important to ensure optimal pubertal development and growth [20].

In terms of gastrointestinal manifestations, patient 1 showed diarrhea, malabsorption and abdominal pain. Chronic diarrhea in APECED can be due to several causes, such as pancreatic exocrine insufficiency, autoimmune enteropathy (AE), lactose intolerance and celiac disease, as well as hypocalcemia [21]. Histological evaluation of the small bowel in typical AE reveals severe enteropathy with atrophy of the villi of the small intestine, crypt hyperplasia and infiltration of mononuclear cells in the lamina propria, while patients with APECED usually do not exhibit those features of AE, such as in our case, and the main or only finding being the loss of EECs and chromogranin A is considered a surrogate marker for these cells [22].

Primary adrenocortical insufficiency, present only in patient 2, most commonly appears after the other two main features, typically between 5 and 15 years of age [1], with symptoms such as fatigue, weight loss, hypotension, salt craving and increased skin pigmentation. Patient 2 showed incidental laboratory findings that suggested adrenal dysfunction, which was then confirmed by 21-hydroxylase antibody positivity with no sign nor symptoms. Therefore, a strict and regular follow-up allowed us to reach a precocious diagnosis, avoiding an adrenal crisis.

Autoimmune hepatitis affects about a fifth of patients and, fortunately, in most cases, it goes away without therapy after a few months [1]. Patient 1 presented with fever and liver enzyme abnormalities, hepatomegaly and ANA and ASMA positivity, but a liver biopsy was not performed for the resolution of signs and symptoms, leading us to consider autoimmune hepatitis as the most probable diagnosis, even without histological confirmation.

Although clinical type 1 diabetes occurs rarely in APECED, elevated titers of GADA are frequently found, as in the two patients presented, and could represent an epiphenomenon of autoimmune insulitis that usually does not lead to clinical manifestation of the disease [23], which is less frequent than in patients with other autoimmune diseases or in healthy children [24].

Patient 2 also had COVID-19 infection with pneumonia, complicated by hypertension, hypokalemia and hypercalcemia. Based on current data, there is no evidence that patients with adrenal insufficiency have an increased risk of contracting COVID-19 but they have a slightly increased overall risk of having infections and an overall increased mortality, possibly explained by an insufficient compensatory increase in the hydrocortisone dose at the time of the onset of an episode of infection. Thus, patients with adrenal insufficiency may be at higher risk of medical complications and eventually at increased mortality risk in the case of COVID-19 infection [25].

There are no significant data describing the effect of COVID-19 on APECED patients, which are certainly not typical examples of adrenal insufficiency, and our case showed different metabolic complications requiring different treatment adjustments, also concerning his chronic therapy for hypoparathyroidism.

The clinical variability of APECED and the absence of a clear genotype–phenotype correlation, also evidenced in our report of two cases with the same *AIRE* genotype but very different phenotypes, could indicate that, in addition to the failure of AIRE function, additional mechanisms may be involved, in a complex interaction between several genetic, epigenetic, immunological and/or environmental factors [26]. Some studies have investigated the effects of additional genetic loci, particularly the human leukocyte antigen (HLA) complex, limited to a few manifestations, but only a weak association has been observed between the HLA type and autoantibody specificities observed in APECED patients [6]. Infectious agents could act as a trigger for a self-reaction through various mechanisms in genetically susceptible subjects, but their role in the clinical expression of APECED has not been demonstrated. Since the *AIRE* gene also appears to be involved in the control of peripheral mechanisms for the maintenance of self-tolerance, a possible role of reduced peripheral tolerance has been hypothesized in the pathogenesis and clinical expression of APECED [26]. A decrease in CD4+CD25+ T regulatory cells (Tregs) has been reported in both adults and children with APECED, but the reduction in circulating Tregs could also be secondary to chronic fungal infection and their pathogenetic role is not defined [6]. Certainly, many factors may play a role in modulating the disease expression in our patients as well, but the current data did not find a definite reason for this variability.

Considering the multi-organ expressiveness of APECED disease, it is of utmost importance that pediatric endocrinologists or endocrinologists, who generally coordinate the multidisciplinary team of providers, have a deep awareness of the multifaceted manifestations and natural history of the disorder to set up a multidisciplinary follow-up, including immunologists, gastroenterologists and eventually other specialists, to promptly recognize and take of care of the development of new disease components or laboratory abnormalities which may appear throughout life [1].

## 4. Conclusions

APECED syndrome is a multisystemic disease, with different combinations of affected organs and autoantibody specificities, that certainly requires a multidisciplinary approach. The genotype does not predict the phenotype, even in siblings with the same mutations, and genetic, epigenetic and environmental factors may play a role. Therefore, further studies might identify new disease-modifying mechanisms or genes influencing the variable expressivity of APECED.

## Figures and Tables

**Table 1 genes-12-00169-t001:** Manifestations of two siblings affected by autoimmune polyendocrinopathy–candidiasis–ectodermal dystrophy (APECED) disease: variability and similarities.

Manifestation	Patient 1, Female, 14 Years Old	Patient 2, Male, 19 Years Old
Hypoparathyroidism	+	+
Primary adrenocortical insufficiency	−	+
Chronic mucocutaneous candidiasis	+	+
Vitiligo	+	−
Alopecia	+	+
Enamel hypoplasia	+	−
Onicodystrophy	+	−
Rosaceiforme dermatisis and acne	−	+
Autoimmune hepatitis	+/−	−
Autoimmune insulitis	+	+
Diarrhea/intestinal malabsorption	+	−
Growth impairment	+	−
Pubertal delay	+	−

## Data Availability

No primary datasets were produced in this study. Data sharing is not applicable to this article.

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
