# Peer review of "Autoimmune Polyendocrinopathy–Candidiasis–Ectodermal Dystrophy in Two Siblings: Same Mutations but Very Different Phenotypes"

_genes, 2021, doi:10.3390/genes12020169_

Round 1

Reviewer 1 Report

The manuscript „ Autoimmune polyendocrinopathy- candidiasis- ectodermal distrophy in two siblings: same mutations byt very different phenotypes”  have typical construction  for case-reports. The main problem described in manuscript: the differences of sign and symptoms between siblings with the AIRE gene mutations interesting .

The Background is to concise. The authors can also use  the  latest publications for example:

Fardi Golyan F, Ghaemi N, Abbaszadegan MR, Dehghan Manshadi SH, Vakili R, Druley TE, Rahimi HR, Ghahraman M. Novel mutation in AIRE gene with autoimmune polyendocrine syndrome type 1. Immunobiology. 2019 Nov;224(6):728-733. doi: 10.1016/j.imbio.2019.09.004. Epub 2019 Sep 6. PMID: 31526676.

Sng J, Ayoglu B, Chen JW, Schickel JN, Ferre EMN, Glauzy S, Romberg N, Hoenig M, Cunningham-Rundles C, Utz PJ, Lionakis MS, Meffre E. AIRE expression controls the peripheral selection of autoreactive B cells. Sci Immunol. 2019 Apr 12;4(34):eaav6778. doi: 10.1126/sciimmunol.aav6778. PMID: 30979797; PMCID: PMC7257641.

In Case reports  the autoimmune hepatitis and hypergonadotropic hipogonadism in patient 1 need the documentation ( hepatic biopsy  result or  autoimmune antibodies documentation, estradiol levels  and LH, FSH levels).

The Disscusion is interesting  but may be  more connected with  APECED manifestations in patients.

The conclusions is compatybile to results and discussion.

The Reference sit is only 11 papers  but lat 2 years  were published over 400 papers about genetics of APECED syndrome.

Author Response

We have expanded the background section, as requested, including also recent literature findings of genetics on APECED syndrome. We have also modified and implemented the discussion, more connected with APECED manifestations in the patients described. We have added further clinical data of our parents, in particular on autoimmune hepatitis and hypogonadism in patient 1, as requested.

Reviewer 2 Report

Carpino and colleagues reported on an atypical presentation of APECED syndrome in two siblings.

The paper is interesting and reports novel insights on the different clinical presentation that APS1 might have.

Minor revisions.

It would be interesting to know:

  • How the genetic analysis was performed (Sanger or Next Generation Sequencing);
  • How the compound heterozygous state was determined: in particular it would need to specify if the pathogenic variants were searched/confirmed in the parents and, in the case, clarify if they were clinically investigated to confirm they are (or not) asymptomatic;
  • It would be interesting to add some more info about the pathogenic variants, if they were identified (in the past) always in autosomal recessive cases or not. This taking into account that the p.(Leu87Pro) was reported only one time in ClinVar.
  • Nomenclature of the pathogenic variants should follows the guidelines at https://varnomen.hgvs.org/: please indicate for both the variants the corresponding aminoacidic change.

Background

Lines 35-36: “…if a sibling is affected or a disease-causing mutation in both AIRE genes”. It seems that APS1 is due to pathogenic variants in more than one gene. Please clarify.

Author Response

We have specified the information on the genetic analysis, as requested, on how it was performed, the parental origin of each variant, and on its pathogenic mechanism, by considering the existing literature data on the same variants. Furthermore, we have expanded the background and the discussion sections, including also recent literature findings of genetics on APECED syndrome.